# ALAS: Active Learning for Autoconversion Rates Prediction from Satellite Data

**Maria Carolina Novitasari[1]**
maria.novitasari.20@ucl.ac.uk

**Johannes Quaas[2,3]**
johannes.quaas@uni-leipzig.de

**Miguel R. D. Rodrigues[1]**
m.rodrigues@ucl.ac.uk

[1]Department of Electronic and Electrical Engineering, University College London
[2]Leipzig Institute for Meteorology, Universität Leipzig
[3]ScaDS.AI - Center for Scalable Data Analytics and AI, Universität Leipzig

## Abstract

High-resolution simulations, such as the ICOsahedral Non-hydrostatic Large-Eddy Model (ICON-LEM), provide valuable insights into the complex interactions among aerosols, clouds, and precipitation, which are the major contributors to climate change uncertainty. However, due to their exorbitant computational costs, they can only be employed for a limited period and geographical area. To address this, we propose a more cost-effective method powered by an emerging machine learning approach to better understand the intricate dynamics of the climate system. Our approach involves active learning techniques by leveraging high-resolution climate simulation as an oracle that is queried based on an abundant amount of unlabeled data drawn from satellite observations. In particular, we aim to predict autoconversion rates, a crucial step in precipitation formation, while significantly reducing the need for a large number of labeled instances. In this study, we present novel methods: custom query strategy fusion for labeling instances – weight fusion (WiFi) and merge fusion (MeFi) – along with active feature selection based on SHAP. These methods are designed to tackle real-world challenges – in this case, climate change, with a specific focus on the prediction of autoconversion rates – due to their simplicity and practicality in application.

## 1 Introduction

Precipitation is a crucial weather and climate phenomenon, with its formation rate being influenced by various factors, including interactions among aerosols, clouds, and precipitation. Understanding these interactions is vital for improving future climate projections, as they represent a major source of uncertainty in estimating climate change's radiative forcing [9].

High-resolution simulations, such as the ICON-LEM [19, 3, 7], offer valuable insights into these interactions. However, it is computationally very expensive. For instance, running ICON-LEM to simulate a single hour of climate data over Germany requires around 13 hours on 300 computer nodes and incurs a cost of approximately EUR 100,000 per day [2]. Given these high costs, it is imperative to seek alternative approaches for understanding complex climate system.

Thus, we propose developing a machine learning (ML) model with active learning (AL) techniques to predict autoconversion rates, a key process in precipitation (rain) formation, which in turn is key to better understanding cloud responses to anthropogenic aerosols [1]. In particular, we propose to use a high-resolution ICON-LEM as an oracle that is queried based on an abundant amount of unlabeled

NeurIPS 2023 AI for Science Workshop.

---

**Algorithm 1** Active Learning with SHAP-Based Feature Selection

---

1: **Input**: $D_{\text{init}}, D, X_{\text{us}}, P, \mathcal{M}, B_{\text{max}}, \mathbf{z} \in \mathbb{R}^p, t$ **Output**: $\hat{\mathcal{M}}, \hat{\mathbf{z}}$: Final model and features
2: $D \leftarrow D_{\text{init}}, \hat{\mathbf{z}} \leftarrow \mathbf{z}, \hat{\mathcal{M}} \leftarrow \emptyset$
3: **while** $|D| \leq B_{\text{max}}$ **do**
4:    **if** $|D| = |D_{\text{init}}|$ or $|D| = \frac{B_{\text{max}}}{2}$ or $|D| = B_{\text{max}}$ **then**
5:       $\hat{\mathcal{M}} \leftarrow \text{train}(\mathcal{M}, D_{\mathbf{z}}); \phi_j = \text{SHAP}(\hat{\mathcal{M}}, \mathbf{z}_j), \forall j; \hat{\mathbf{z}} \leftarrow \mathbf{z} \setminus \{j : |\phi_j| < t\}$
6:    **end if**
7:    $\hat{\mathcal{M}} \leftarrow \text{train}(\mathcal{M}, D_{\hat{\mathbf{z}}})$
8:    $P \leftarrow \text{Active Learning Step}(\hat{\mathcal{M}}, X_{\text{us}})$
9:    Ask oracle to label points in $P$; $D \leftarrow D \cup P$; $X_{\text{us}} \leftarrow X_{\text{us}} \setminus x_i : x_i \in P$
10: **end while**
11: **return** $\hat{\mathcal{M}}, \hat{\mathbf{z}}$

---

data drawn from satellite data. Our aim with active learning is to minimize the number of labeled instances required to train the machine learning model. We will demonstrate that active learning allows us to achieve greater accuracy with fewer labeled data points by selecting the most valuable instances from a pool of unlabeled data, thus reducing overall costs.

Several AL algorithms have attempted to combine both informativeness and representativeness measures when selecting optimal query instances, primarily in the context of classification problems [4, 8]. [18] introduced an approach that strives to maximize diversity. Our method draws inspiration from the principles of combining informativeness, representativeness, and diversity, akin to the approach undertaken by [6] and [12]. However, our method is specifically tailored for regression problems, setting it apart from the aforementioned classification-focused.

Our research contributes to the field in several significant ways. First, to the best of our knowledge, we are the first to apply AL in the field of high-resolution climate modeling, specifically within the context of the very expensive ICON-LEM simulation, with a specific focus on the autoconversion process – a process by which cloud droplets grow larger and transform into raindrops. Secondly and thirdly, we introduce active feature selection using SHAP (SHapley Additive exPlanations), and innovative query strategy fusion techniques: query strategy fusion by weight (WiFi) and query strategy fusion by merging (MeFi) which are straightforward and convenient in practice.

## 2 Proposed Methods

We introduce active feature selection using SHAP and novel query strategies that consider three crucial factors when choosing unlabeled instances in AL: informativeness, representativeness, and diversity, explained in the following subsections. For our discussion, let the following notations be defined: $D_{\text{init}}$ as the initial labeled data, $D$ as the current labeled data, $X_{\text{us}}$ as the small unlabeled pool, $P$ as the set of points to be labeled, $\mathcal{M}$ as the ML model, $B_{\text{max}}$ as the maximum budget (number of labeled), $\mathbf{z} \in \mathbb{R}^p$ as the full feature vector, and $t$ as the SHAP threshold.

**Active Feature Selection**   Our approach employs SHAP to assess feature contributions, eliminating insignificant features throughout certain AL stages (see Alg. 1).

**Informativeness**   Given a Gaussian process regression model $f \sim \mathcal{GP}(m, k)$ where $m$ is the prior mean function and $k$ is the prior covariance kernel, the predictive distribution at a new input $x_*$ is Normal with mean $\mu(x_*)$ and variance $\sigma^2(x_*)$. In informativeness-based sampling with Gaussian Process Regression (GPR) [17], we leverage the model's predictive standard deviation, denoted as $l_{\text{inf}}$, to quantify prediction uncertainty. Our goal is to choose $P$ data points for labeling that have the highest $l_{\text{inf}}$ values, as these points correspond to regions where the model is least certain. The details of our informativeness-based sampling algorithm are outlined in Appendix B1.

**Representativeness**   In this section, we introduce a straightforward approach that involves selecting a number of $|P|$ data points to label based on the most representative values they hold (i.e., those closest to their centroid cluster), denoted as $l_{\text{rep}}$, as a query strategy in AL regression. The

optimal number of clusters is determined using the Silhouette method on $X_{\text{us}}$. The details of our representativeness-based sampling algorithm are outlined in Appendix B2.

**Diversity** In diversity-based sampling, we select $P$ data points that maximize dissimilarity within their clusters, denoted as $l_{\text{div}}$. By calculating the average dissimilarity for each data point within its cluster, we identify those that contribute the most to dataset diversification. The optimal number of clusters is determined using the Silhouette method on $X_{\text{us}}$. The details of our diversity-based sampling algorithm are outlined in Appendix B3.

**Weight Fusion (WiFi)** We propose the Weight Fusion (WiFi) query strategy, with $\alpha$ and $\beta$ as weight trade-offs. $\alpha$ governs informativeness vs. representativeness, while $\beta$ manages the trade-off between informativeness-representativeness and diversity. Higher $\alpha$ values emphasize representativeness, and higher $\beta$ values prioritize diversity. WiFi is defined as:

$$\text{WiFi}(x_*) = (1 - \beta)\left((1 - \alpha) \cdot l_{\text{inf}}(x_*) + \alpha \cdot l_{\text{rep}}(x_*)\right) + \beta \cdot l_{\text{div}}(x_*)$$

where $x_* \in X_{\text{us}}$. Details of $l_{\text{inf}}, l_{\text{rep}}, l_{\text{div}}$ are explained in the previous subsections, where they denote informativeness, representativeness, and diversity scores. We select the top $P$ points in $X_{\text{us}}$ based on their descending WiFi rank and optimize $\alpha$ and $\beta$ using initial labeled data.

**Merge Fusion (MeFi)** MeFi is a novel query strategy that optimally balances informativeness, representativeness, and diversity by merging the top $\frac{|P|}{3}$ data points from each category ($L_{\text{inf}}, L_{\text{rep}}, L_{\text{div}}$), defined as follows:

$$\text{MeFi} = \frac{|P|}{3}L_{\text{inf}} \cup \frac{|P|}{3}L_{\text{rep}} \cup \frac{|P|}{3}L_{\text{div}}$$

## 3   Experimental Results

### 3.1   Datasets

We trained and validated our models using ICON-LEM output over Germany on May 2, 2013, from 09:55 am to 1 pm local time. The test dataset consists of two subsets: one covering the entire Germany region on May 2, 2013, at 13:20 local time, and another encompassing the North Atlantic region on September 1, 2014, at 13:00 local time. As for the satellite observation data, we use cloud product level-2 of Terra and Aqua MODIS [14, 15]. Details of our datasets, including various testing scenarios, are provided in Appendix A and C.2.1.

### 3.2   Active Learning (AL)

**Initial Active Learning Settings** We utilized a pool-based AL regression approach with a large training pool of about 4 million unlabeled data points and a large validation pool of approximately 900,000 data points. We conducted 100 experiments – including active feature selection, cluster number selection, AL, and $\alpha$ and $\beta$ hyperparameter tuning – and averaged the results. In each experiment, we sampled small training ($X_{\text{us}}$) and validation pools of 1,000 and 250 data points, respectively, with $|D_{\text{init}}| = 10$ and $|P| = 3$. We employed GPR to train our ML models. Our initial model takes the cloud effective radius (CER) and pressure (P), parameters of the cloud microphysical state typically obtained from satellite retrievals, as input. The model output is the autoconversion rates derived from ICON-LEM.

**Active Feature Selection** In this step, we selected our features using the active feature selection algorithm explained in Section 2. Our results highlight CER as the most influential feature in predicting autoconversion rates, while the contribution of P is relatively small, as shown in Fig. 1. We validated our results by performing Gaussian process regression across different sample sizes (10, 50, and 100) and evaluating the outcomes. Consistently, the results show that using P alone outperforms using both P and CER as input features (see details in Appendix C.1.1).

**Selection of the Number of Clusters, Alpha, and Beta** We determined the optimal number of clusters using the Silhouette method on $X_{\text{us}}$. The best number of clusters was found to be 2. The results for $\alpha$ selection using initial data points are illustrated in Fig. 2. The optimal $\alpha$ value is determined to be 0.5, signifying an equilibrium between 50% informativeness and 50%

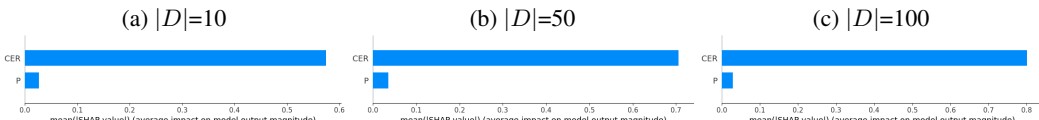

Figure 1: The results of active feature selection with SHAP.

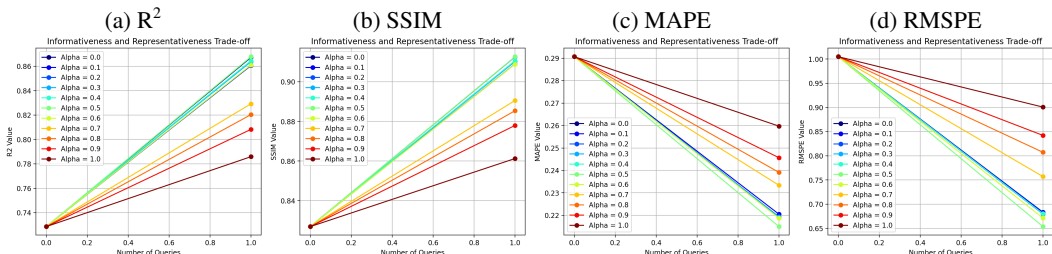

Figure 2: Exploring the alpha trade-off: balancing informativeness and representativeness.

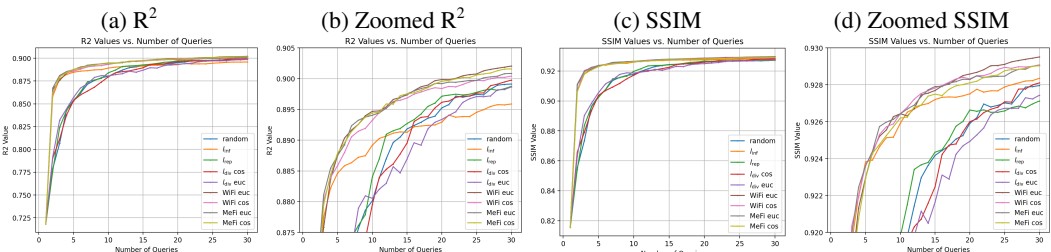

Figure 3: Evaluation of different query strategies in active learning with $R^2$ and SSIM.

representativeness. The optimal $\beta$ value for diversity based on Euclidean distance is 0.4, resulting in a balanced combination of 40% informativeness-representativeness and 60% diversity, while for inverse cosine-based diversity, it is identified as 0.5 (see Appendix C.1.2 for details on the selection of $\beta$).

**Active Learning Results**  We assess the AL query strategy performance using $R^2$ and SSIM metrics, shown in Fig. 3. $R^2$ indicates that $l_{\text{inf}}$, WiFi, and MeFi (Euclidean (euc); inverse cosine (cos)) achieve faster convergence than random (baseline), $l_{\text{rep}}$, and $l_{\text{div}}$. However, $l_{\text{inf}}$ eventually lags behind others. WiFi and MeFi consistently outperform baseline and individual aspects ($l_{\text{inf}}$, $l_{\text{rep}}$, $l_{\text{div}}$) across all query iterations. SSIM results closely align with the $R^2$ findings, showing that $l_{\text{inf}}$, WiFi, and MeFi, consistently outperform others, with WiFi and MeFi still maintaining their lead. WiFi, in particular, excels when using the Euclidean metric for both $R^2$ and SSIM.

### 3.3 Autoconversion Rates Prediction

We employ GPR with an RBF and white noise kernel to train our model. To determine the optimal hyperparameters for the kernel, we employ random search cross-validation. Our training dataset consists of only 100 labeled data points selected using the best AL query strategy explained in the previous subsection (WiFi Euclidean), while we reserve 250 data points for validation. This represents <0.01% of the total actual labeled data available and only 47% of the labeled data needed by the baseline (Appendix C.1.3). We utilize a significantly smaller amount of data in comparison to the work by [13], who utilized the entire cloud-top dataset. For the input, we use CER as determined by our previous experiment using SHAP.

**Simulation Model (ICON)**  We test our model on simulation data in 3 different scenarios: (1) ICON-LEM Germany (different times), (2) Cloud-top ICON-LEM Germany (satellite-like data), and (3) Cloud-top ICON-NWP Holuhraun (different data, time, location, and resolution), details in Appendix C.2.1. The results in Table 1 demonstrate that SSIM values exceed 90% for all scenarios, with scenarios 1 and 2 also achieving over 90% for $R^2$. Scenario 3, despite using different data

Table 1: Evaluation of autoconversion prediction results on three different testing scenarios.

| Testing Set | $R^2$ | MAPE | RMSPE | SSIM | PSNR (dB) |
|---|---|---|---|---|---|
| 1 | 90.18% | 9.31% | 12.19% | 90.52% | 26.14 |
| 2 | 90.32% | 10.35% | 13.20% | 90.29% | 26.09 |
| 3 | 85.09% | 8.33% | 22.47% | 91.66% | 26.01 |

(a) CER

(b) Aut

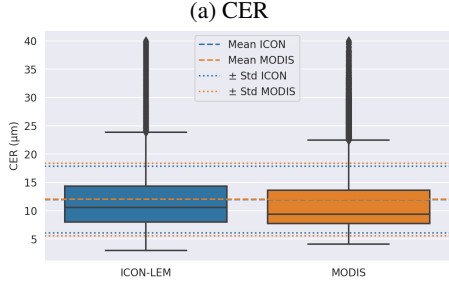
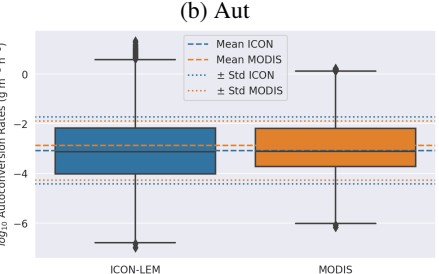

Figure 4: Mean, standard deviation (Std), median, and percentiles (p25, p75) of cloud-top ICON-LEM and MODIS variables over Germany: cloud effective radius (CER) and autoconversion rates (Aut).

in terms of time, location, and resolution, still achieves an $R^2$ slightly above 85%. These findings highlight the model's capability to accurately estimate autoconversion rates when utilizing model-simulated satellite data, without the need for further adjustments like fine-tuning. This minimizes the need for additional data collection and time-consuming training processes. Visual representation included in Appendix C.2.2.

**Satellite Observation (MODIS)**    This experiment aims to assess our model's ability to predict autoconversion rates using real satellite data, specifically by testing the model on such data. We focused on comparing the autoconversion rate predictions from the MODIS satellite with cloud-top ICON simulation output over Germany (latitude: 47.50° to 54.50° N, longitude: 5.87° to 10.00° E). While it is worth noting that direct comparisons between satellite predictions and simulation models cannot be made directly due to differences in cloud placement, Fig. 4 demonstrates that the MODIS autoconversion rate predictions statistically align with those from cloud-top ICON-LEM Germany. The mean, standard deviation, median, and percentiles of autoconversion rates demonstrate a close agreement. It shows that autoconversion rates can be well estimated from satellite-derived CER data using our method.

## 4    Conclusions

In this study, we have provided a computationally efficient solution for understanding the key process of precipitation formation, specifically the autoconversion process. This process plays a crucial role in advancing our understanding of how clouds respond to anthropogenic aerosols [11], and ultimately, climate change. Importantly, we have shown it is possible to predict autoconversion rates accurately using less than 0.01% of the expensive labeled data from high-resolution ICON-LEM simulation. Our machine learning model achieves good performance on both atmospheric simulation and satellite data, while requiring only 47% of the data needed by the baseline strategy. This demonstrates a cost-effective approach to train an accurate model with minimal labeled data. Additionally, we introduced innovative techniques: custom query strategies for active learning, WiFi and MeFi, along with active feature selection using SHAP. These methods were specifically designed to address real-world problems due to their practical simplicity. Our custom query strategy fusion, WiFi and MeFi, consistently outperformed the baseline query strategy, as well as the individual aspects of informativeness, representativeness, and diversity. For simplicity, we used only the initially selected data points for hyperparameter selection in this work, but exploring an adaptive method for selecting hyperparameters in the WiFi query strategy could be a potential direction for future research.

## Acknowledgments and Disclosure of Funding

We would like to express our sincere appreciation to the anonymous reviewers for their valuable feedback. This research receives funding from the European Union's Horizon 2020 research and innovation programme under Marie Skłodowska-Curie grant agreement No 860100 (iMIRACLI). This work used resources of the Deutsches Klimarechenzentrum (DKRZ) granted by its Scientific Steering Committee (WLA) under project ID bb1143. The model output data used for the development of the research in the frame of this scientific article is available on request from in tape archives at the DKRZ, which will be accessible for 10 years.

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

## A  Dataset

We use datasets from ICON-LEM output from a simulation of the conditions over Germany on 2 May 2013, where distinct cloud regimes occurred, allowing for the investigation of quite different elements of cloud formation and evolution [7]. We study a time period of 09:55 UTC to 13:20 UTC, corresponding to the polar-orbiting satellite overpass times. Our focus is on ICON-LEM with a 156 m resolution on the native ICON grid, then regridded to a regular 1 km resolution to match the resolution of MODIS.

The autoconversion rates in our training and testing data were derived using the two-moment microphysical parameterization of Seifert and Beheng (2006). The autoconversion rates for cloud tops that simulate satellite data were determined by selecting rates where the cloud optical thickness, calculated from top to bottom, exceeds 1. The optical thickness represents the extent to which optical satellite sensors can retrieve cloud microphysical information.

We use dataset of ICON numerical weather prediction (ICON-NWP) Holuhraun which were performed over Holuhraun volcano for a week from 1 September to 7 September 2014 to further test the performance of our machine learning models [10, 5]. The dataset has a horizontal resolution of approximately 2.5 km. As for the satellite observation data, we use cloud product level-2 of Terra and Aqua MODIS [14, 15].

## B  Proposed Methods

We introduce novel query strategies that take into consideration three crucial factors when selecting unlabeled instances in active learning: informativeness ($l_{\text{inf}}$), representativeness ($l_{\text{rep}}$), and diversity ($l_{\text{div}}$). Due to page limitations, we include the details of each category ($l_{\text{inf}}$, $l_{\text{rep}}$, and $l_{\text{div}}$) of the query strategy in this appendix section.

### B.1  Informativeness

Our informativeness-based (uncertainty) sampling active learning query strategy is shown in Algorithm B1.

---
**Algorithm B1** Informativeness-based Sampling

---
1: **Input**: Small unlabeled pool $X_{\text{us}}$, GP model $f \sim \mathcal{GP}(m, k)$ **Output**: $P$ points to label
2: $l_{\text{inf}} \leftarrow \emptyset$
3: Use GP to compute $\mu(x_*), \sigma^2(x_*)$ for all $x_* \in X_{\text{us}}$
4: **for** each $x_* \in X_{\text{us}}$ **do**
5:     Compute predictive std $\sigma(x_*)$.
6:     Set Informativeness score $l_{\text{inf}}(x_*) = \sigma(x_*)$
7: **end for**
8: Normalize $l_{\text{inf}}$ to $[0, 1]$
9: $\hat{X} \leftarrow$ indices of top $P$ points in $X_{\text{us}}$ ranked in descending order by $l_{\text{inf}}$
10: **return** $\hat{X}$ (Indices of $P$ points to query)

---

### B.2  Representativeness

The algorithm for our representativeness-based sampling is outlined in Algorithm B2.

### B.3  Diversity

Our diversity-based sampling is shown in Algorithm B3.

## C  Experimental Results

### C.1  Active Learning

#### C.1.1  Active Feature Selection

Initially, we started with two candidate features because not all variables in the ICON-LEM output align with satellite data. Consequently, we narrowed our selection to inputs typically derived from satellite retrievals, which limited us to two variables: cloud effective radius (CER) and pressure (P). While we acknowledge the existence of other potential features, such as liquid water path (LWP) and cloud optical thickness (COT), these variables

---

**Algorithm B2** Representativeness-based Sampling

---
1: **Input**: Small unlabeled pool $X_{us}$ **Output**: $P$ points to label
2: $l_{rep} \leftarrow \emptyset$
3: Perform $k$-means clustering on $X_{us}$, where $k$ is determined using the Silhouette method.
4: **for** each $x_* \in X_{us}$ **do**
5:     Compute $d(x_*, c_i)$ where $c_i$ is the centroid of the cluster containing $x_*$.
6:     Set Representativeness score $l_{rep}(x_*) = \frac{1}{d(x_*, c_i)}$
7: **end for**
8: Normalize $l_{rep}$ to $[0, 1]$
9: $\hat{X} \leftarrow$ indices of top $P$ points in $X_{us}$ ranked in descending order by $l_{rep}$
10: **return** $\hat{X}$ (Indices of $P$ points to query)

---

---

**Algorithm B3** Diversity-based Sampling

---
1: **Input**: Small unlabeled pool $X_{us}$ **Output**: $P$ points to label
2: $l_{div} \leftarrow \emptyset$
3: Perform $k$-means clustering on $X_{us}$, where $k$ is determined using the Silhouette method.
4: **for** each $x_* \in X_{us}$ **do**
5:     Let $C_i$ be the cluster containing $x_*$
6:     Compute $\bar{d}(x_*) = \frac{1}{|C_i|} \sum_{x_j \in C_i} d(x_*, x_j)$ where $d(\cdot, \cdot)$ is a dissimilarity measure (e.g., Euclidean distance, reverse cosine similarity).
7:     Set Diversity score $l_{div}(x_*) = \bar{d}(x_*)$
8: **end for**
9: Normalize $l_{div}$ to $[0, 1]$
10: $\hat{X} \leftarrow$ indices of top $P$ points in $X_{us}$ ranked in descending order by $l_{div}$
11: **return** $\hat{X}$ (Indices of $P$ points to query)

---

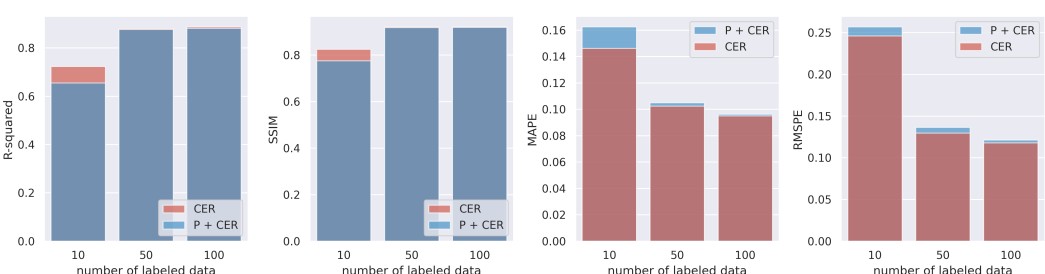

Figure C1: Active Feature Selection. Pressure (P), Cloud Effective Radius (CER).

are vertically integrated and do not provide information per layer. Therefore, we did not include them in our current analysis. However, future research directions may involve, for example, predicting COT per layer as part of our ongoing research.

We validated our results by performing Gaussian process regression across different sample sizes (10, 50, and 100) and evaluating the outcomes. Consistently, the results show that using P alone as input features is better than using both P and CER, as illustrated in Figure C1.

### C.1.2 Selection of Beta

Figure C2 illustrates the selection of $\beta$ using Euclidean distance metrics, while Figure C3 showcases the results of $\beta$ selection with inverse cosine similarity applied to the initial data points $D_{init}$.

### C.1.3 Active Learning Results

Figure C4 illustrates the label efficiency of our approach compared to the baseline, quantifying how much less labeled data is needed to achieve similar results based on the best achievable results using the random query strategy. It demonstrates that, on average, our best query strategy (WiFi Euclidean) requires only 47% of the

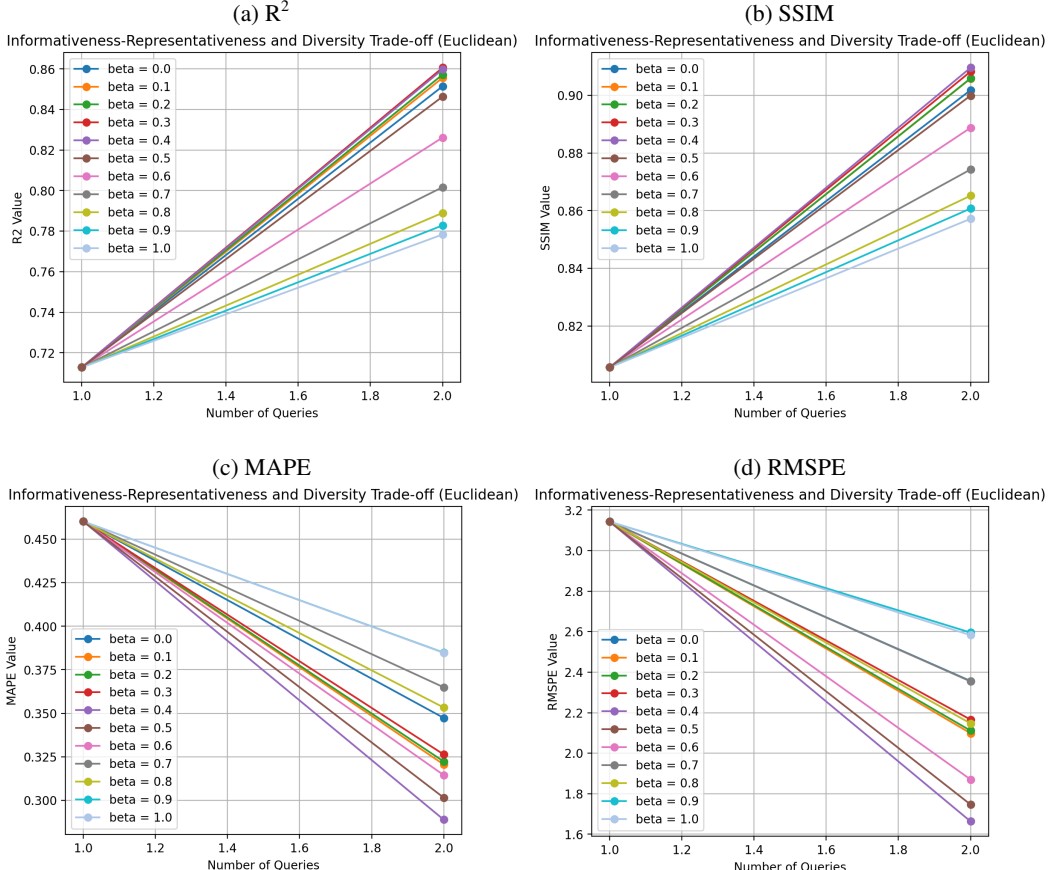

Figure C2: Exploring the beta trade-off: balancing informativeness-representativeness and diversity (Euclidean distance).

labeled data to reach comparable results. Specifically, we need 64% ($R^2$), 53% (SSIM), 40% (RMSE), and 32% (RMSPE) of the labeled data, relative to the baseline (100%), to obtain similar outcomes for different metrics.

## C.2 Autoconversion Rates Prediction

### C.2.1 Testing Datasets/Scenarios on Simulation Model

We evaluate our final machine learning model using different testing datasets and scenarios associated with the ICON-LEM simulations over Germany and the ICON-NWP simulations over Holuhraun, as follows:

1. *ICON-LEM Germany*: In this testing scenario, we evaluate the performance of our machine learning models using the same data that was utilised during its training process. This data, which consists of a set of cloud effective radius and autoconversion rates, was collected through the use of ICON-LEM simulations specifically over Germany. The testing data, however, differs from the training data as we focus on a different time period, specifically 2 May 2013 at 1:20 pm. This approach enables us to assess the model's generalisation capability to new data within the same region and day, while considering significant weather variations that evolved considerably [7]. Number of data points: approximately 950,000.

2. *Cloud-top ICON-LEM Germany*: In this testing scenario, we evaluate the performance of our machine learning model by utilising the same data as in the previous scenario, with the exception that we are only considering the cloud-top information of the data. We extract this cloud-top 2D data from the 3D atmospheric simulation model by selecting the variable value at any given latitude and longitude where the cloud optical thickness exceeds 1, integrating vertically from cloud-top. Number of data points: approximately 144,000.

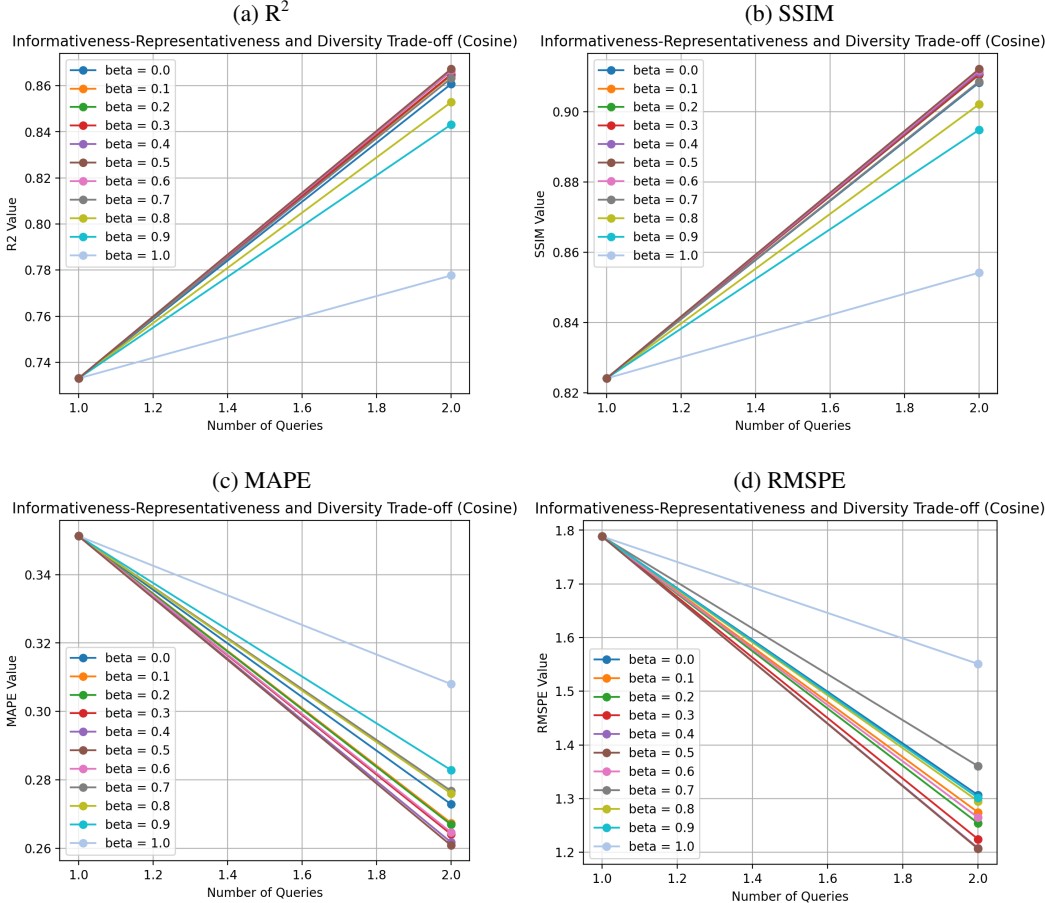

Figure C3: Exploring the beta trade-off: balancing informativeness-representativeness and diversity (inverse cosine similarity).

3. *Cloud-top ICON-NWP Holuhraun*: This final testing scenario uses distinct data from that of previous scenarios. In particular, we use cloud-top of ICON-NWP Holuhraun data that was acquired at a different location, time, and resolution compared with the data used in the previous scenarios. The ability of the model to perform well in the presence of new data is important in many practical applications, allowing the model to make accurate predictions on unseen data, adapting to varying geographical locations, and adapting to different metereological conditions. Number of data points: approximately 1.7 million.

The performance of each model is evaluated by calculating a range of metrics, including $R^2$, MAPE (Mean Absolute Percentage Error), RMSPE (Root Mean Squared Percentage Error), Peak Signal-to-Noise Ratio (PSNR), and Structural Similarity Index (SSIM), on the testing data. Prior to calculating each metric, the data is normalised by transforming it using base 10 logarithms and then scaling it to a range between 0 and 1.

### C.2.2 Simulation Model (ICON)

The visual representation of autoconversion rate predictions for ICON-LEM Germany and ICON-NWP Holuhraun under various testing scenarios can be seen in Figure C5. These figures demonstrate our model's ability to accurately capture and reproduce key groundtruth features. This is evident in the strong resemblance between the groundtruth and our model's predictions, which show minimal deviations, generally below 20% and predominantly around less than 10%. In summary, these results confirm our model's effectiveness in diverse scenarios, including atmospheric simulations and satellite-like data, with a high degree of accuracy.

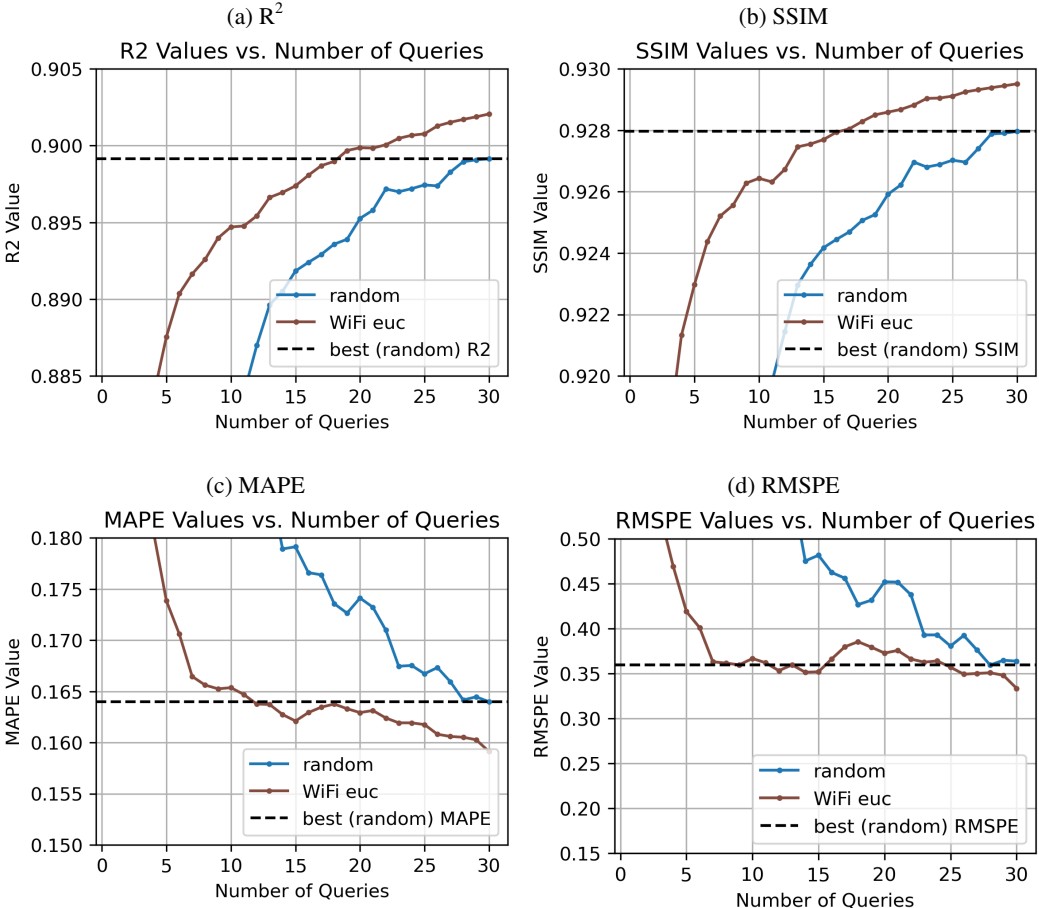

Figure C4: Label efficiency comparison: The figure demonstrates how many labeled data points are needed to achieve comparable results across multiple metrics when using the best query strategy (WiFi Euclidean) compared to the random (baseline) query strategy.

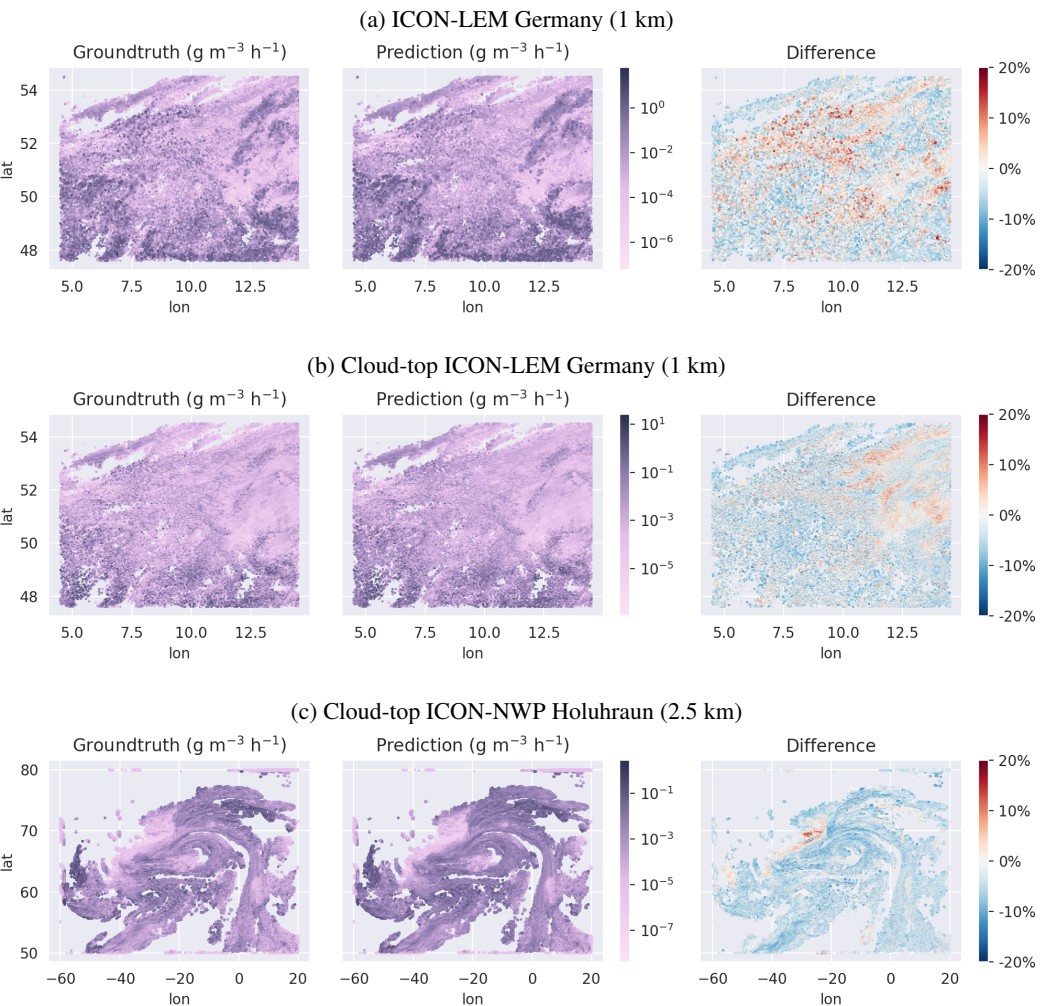

Figure C5: Visualization of the autoconversion prediction results of ICON-LEM Germany and ICON-NWP Holuhraun. The left side of the image depicts the groundtruth, while the middle side shows the prediction results obtained from the GP model. The right side displays the difference between the groundtruth and the prediction results. The top image (a) compares the groundtruth and predictions from ICON-LEM Germany at a resolution of 1 km, while the second image (b) focuses on cloud-top information only at a resolution of 1 km. The third figure (c) illustrates the comparison between groundtruth and predictions of the ICON-NWP Holuhraun data with a horizontal resolution of 2.5 km, focusing on cloud-top information only.

