# OpenReview forum: "ALAS: Active Learning for Autoconversion Rates Prediction from Satellite Data"
_NeurIPS.cc/2023/Workshop/AI4Science — NeurIPS2023-AI4Science Poster_

### Official Review · Reviewer_ah1B · 2023-10-12
**Official Comments**

**Rating:** 6
**Confidence:** 3

**Review:**

### Paper Summary

This paper focuses on the development of cost-efficient solution for understanding the key process of precipitation formation. The authors propose four novel techniques, including custom query strategy fusion, WiFi, MeFi, and SHAP-based feature selection, to predict autoconversion rates. Extensive quantitative and qualitative experimental results demonstrate that this proposed approach consistently delivers excellent performance in autoconversion rate prediction while keeping computational overhead to a minimum.



### Strengths

1. This paper addresses the challenge of understanding the climate system using high-resolution simulations in a cost-effective manner. This is an important topic in climate science.
2. The proposed active learning techniques and query strategies provide a practical and efficient way to predict autoconversion rates and reduce the need for labeled instances.
3. The paper introduces the use of SHAP-based feature selection, which is novel and tailored for regression problems.
4. The experimental assessment is solid, encompassing meticulous implementation and parameter configuration, an evaluation of both quantitative and qualitative performance, as well as essential ablation studies.



### Weaknesses

1. The paper lacks comparisons and discussions with widely known baselines in the field. It would be valuable to compare the proposed method with other state-of-the-art approaches to highlight its advantages and limitations.



### Questions:

1. Have you conducted any comparisons with existing approaches in the field? It would be valuable to see how the proposed method performs compared to other methods for predicting autoconversion rates.

2. Can you provide more insights into the limitations of the proposed method? Are there any scenarios or conditions where the method may not perform well?

---

### Official Review · Reviewer_mVXZ · 2023-10-23
**Active learning with Gaussian process regression applied to scalable autoconversion rate prediction from simulated and satellite observations**

**Rating:** 7
**Confidence:** 3

**Review:**

This proposal presents an active learning approach with Gaussian process regression (GPR) to speed-up simulator prediction of auto-conversion rate of precipitation process. Active learning is based on heuristics with feature selection, model uncertainty, and clustering-based query strategies. Proposed approaches are experimentally evaluated on ICON-LEM simulations and satellite observations from MODIS. The structure and presentation of the paper are clear, and the proposed active training point selection heuristics and their properties are evaluated adequately, showing usefulness and effectiveness of using only very small number of training examples. Some of the analysis could be more detailed, e.g., computational costs savings of GPR model against simulator. Furthermore, the given forecasting problem is very low-dimensional (only two input features) to fully show the capabilities of chosen feature selection strategies.

Pros
- New application of active learning
- Some novel algorithms and heuristics for choosing the training points
- Work in practice with very small number of labelled data points

Cons
- No detailed analysis of active learning benefits from computational cost perspectives (some rough numbers of simulation computation
costs described in intro)
- Missing ICON-LEM vs. GPR model (i.e., pre-processing-training-evaluations) comparison of computing times
- Very low-dimensional datasets to shown the usefulness of proposed algorithms (e.g., active feature selection)
- Lack of baseline against GPR model

Some minor suggestions:
- Figure 2: could be more clear if each  individual and feature combination would be presented by own bar plot (in each subfigure)

---

### Meta-Review · Area_Chair_cj8r · 2023-10-27

**Recommendation:** Accept (Poster)
**Confidence:** 2

**Metareview:**

Both reviewers concur on the problem's significance, the effectiveness of the proposed active learning method, and the strong empirical results. I encourage the authors to consider the reviewer's suggestions in the following revision.